# Mechanism for Adsorption, Dissociation, and Diffusion of Hydrogen in High-Entropy Alloy AlCrTiNiV: First-Principles Calculation

**DOI:** 10.3390/nano14171391

**Published:** 2024-08-26

**Authors:** Weilong Zheng, Liangliang Wu, Qilin Shuai, Zhaoqiang Li, Haoqi Wang, Wei Fu, Zhenxiong Jiang, Chuang Zhao, Qingsong Hua

**Affiliations:** 1School of Physics and Astronomy, Beijing Normal University, Beijing 100091, China; 202221220009@mail.bnu.edu.cn (W.Z.); 202131220021@mail.bnu.edu.cn (Q.S.); 202121220025@mail.bnu.edu.cn (W.F.); 202221220016@mail.bnu.edu.cn (Z.J.); 202321220017@mail.bnu.edu.cn (C.Z.); 2Laboratory of Beam Technology and Energy Materials, Advanced Institute of Natural Sciences, Beijing Normal University, Zhuhai 519087, China; llwu@bnu.edu.cn; 3Radiation Technology Institute, Beijing Academy of Science and Technology, Beijing 100875, China; wanghq@bnu.edu.cn

**Keywords:** DFT, HEA, hydrogen adsorption, hydrogen diffusion, energy barrier, zero-point energy

## Abstract

To investigate hydrogen behaviors in the high-entropy alloy AlCrTiNiV, density functional theory and transition state theory were used to explore the molecular $H_{2}$ absorption and dissociation and the atomic H adsorption, diffusion, and penetration progress. The $H_{2}$ molecule, where the H-H band is parallel to the surface layer, is more inclined to absorb on the top site of the Ti atom site of first atomic layer on the AlCrTiNiV surface, then diffuse into the hollow sites, through the bridge site, after dissociating into two H atoms. Atomic H is more likely to be absorbed on the hollow site. The absorption capacity for atomic H on the surface tends to decline with the increase in H coverage. By calculating the energy barriers of atomic H penetration in AlCrTiNiV, it was indicated that lattice distortion may be one important factor that impacts the permeation rate of hydrogen. Our theory research suggests that high-entropy alloys have potential for use as a hydrogen resistant coating material.

## 1. Introduction

In recent years, hydrogen energy has increasingly become a substitute for fossil fuels. Inevitably the storage of hydrogen will become a significantly important issue for us. However, the internal lattice structure of the material could be altered if atomic H gets inside it. In some severe cases, the phenomenon of hydrogen embrittlement can be discovered [1,2,3,4]. Depositing hydrogen-permeation barrier (HPB) coatings on the material’s surface is a common and effective way to prevent hydrogen from entering the material and destroying the stability of the structure. Examples such as Al_2_O_3_/Er_2_O_3_ [5,6], CrN/TiN [7,8], and TiC/SiC [9,10] composite coatings, have exhibited an excellent ability to resist hydrogen isotopic permeation. However, coatings with different compositions, thicknesses, and defect densities can lead to significant variations in the permeation reduction factors (PRFs) of barriers between 10 and 10,000 according to our literature research [11]. What is the main factor influencing PRFs and how does it control the size of the PRFs?

The process of hydrogen diffusion through materials involves several steps: H_2_ molecule adsorption on the surface, dissociation into H atoms, atomic H permeation into materials, diffusion through the crystal structure, and recombination into H_2_ molecules on the other side of the coating. The hydrogen resistance of materials hinges on the combined activation energies and the physical interactions between the hydrogen-blocking material and hydrogen. To explore the physics mechanisms of the interactions between hydrogen and barrier materials, Hollenberg et al., considering such hydrogen isotopic behaviors with barriers and the dependence of the permeation rate on the partial pressure of the hydrogen isotope, proposed the composite diffusion model, the area defect model, and the surface desorption model [12]. These models reveal that the PRFs are controlled by the hydrogen molecule dissociation rate on the barrier surface or by the hydrogen atom penetration rate within the barrier. However, despite a significant amount of research, there is still not a unified statement on the microphysical mechanism of hydrogen permeation. It is exceedingly necessary to move forward a step to study the theory of the microscopic mechanisms involved in the interactions between hydrogen and hydrogen-resistant materials in order to better understand the functional mechanisms of HPB materials and explore advanced HPBs.

In recent years, high-entropy alloys (HEAs) have attracted significant attention due to their excellent mechanical properties [13,14]. HEAs are usually composed of five or more metal and nonmetal elements in equimolar or nearly equimolar ratios. One prominent and notable impact of HEAs is their high entropy effect. The stability of HEAs depends on their entropy levels. According to the equation ΔG = ΔH − TΔS, reducing Gibbs free energy enhances alloy stability. The severe lattice distortion (SLD) effect can change a material’s lattice structure and then may restrict hydrogen diffusion through the material. Co, Cr, Mn, Fe, and Ni usually help slow down diffusion, a phenomenon known as the sluggish diffusion effect. The cocktail effect is also an important peculiarity of HEAs. This effect alters material magnetization, flexibility, strength, coercivity, and electrical conductivity, which can be used to design the desired performance of an HEA.

In addition, among all materials with excellent hydrogen resistance performance, the elements Ti, Al and V have been found to be the most relevant among the metal elements that are found in materials, such as TiN [15], TiAlN [16], TiMoN [17], VC [18], etc. In this study, we aim to investigate the high-entropy alloy AlCrTiNiV, composed of an equal atomic ratio of these five elements, which may have good hydrogen resistance performance. There is a large range in the atomic radii of the five elements from 118 pm to 206 pm, which may result in a severe SLD effect. Density functional theory (DFT) calculations are most commonly employed to predict an important physical property of solids, namely their crystal structure, and can be used to determine the diffusion mechanisms of hydrogen atoms in metals, alloys, and ceramics [19,20,21]. Transition state theory (TST) provides a comprehensive framework for studying and understanding the mechanisms and rates of chemical reactions. In this study, DFT provides the detailed energetic and electronic information necessary to identify possible reaction pathways and diffusion mechanisms, while TST translates this information into kinetic predictions that are critical for practical applications. By applying TST to the processes of hydrogen adsorption, dissociation, and diffusion in high-entropy alloys, we can gain valuable insights into the behavior of hydrogen in these materials. This understanding is crucial for developing efficient hydrogen-permeation barriers and optimizing the performance of high-entropy alloys in hydrogen-related applications [22]. DFT calculations can provide extensive microscopic information about systems that cannot be obtained through experiments alone and can further guide the design of more comprehensive experimental schemes. In this study, we employ these two methods to investigate the diffusion behaviors of hydrogen in the high-entropy alloy AlCrTiNiV at elevated temperatures and hope to gain valuable insights.

## 2. Computational Method and Model

All the DFT calculations were performed using the Vienna Ab initio Simulation Package (VASP) [23,24,25]. Considering that nickel is a magnetic element, spin polarization was turned on. The electron wave functions were described using the projected augmented-wave (PAW) [26] method, and the exchange–correlation effects were treated using the Perdew-Burke-Ernzerhof (PBE) [27] functional under the generalized gradient approximation (GGA) [28]. In this study, the HEA AlCrTiNiV consists of five metal elements in equal atomic ratios, indicating that the total number of atoms in the HEA model should be a multiple of five. To balance computational accuracy and efficiency, a 5 × 2 × 3 BCC supercell consisting of 60 atoms was used for the HEA AlCrTiNiV. The structural model of BCC AlCrTiNiV was built using special quasi-random structure (SQS) [29] method with Alloy Theoretic Automated Toolkit (ATAT) [30]. It is worth mentioning that FCC and HCP were also built and optimized. However, the energies of both were ultimately unstable. Therefore, BCC was chosen as the final structure. As shown in Figure 1a, for convenience of calculation, the (100) surface of (3 × 2) BCC AlCrTiNiV was cut and the slab model, consisting of 10 atomic layers with a total of 60 atoms, was constructed, and repeated periodically in both the x and y dimensions. The top five layers are relaxed, while the other five layers are frozen. The vacuum layer width of 1.5 nm between slab models in the (100) direction was proved to be wide enough to ensure that the adjacent slabs are independent and unaffected. It has also been shown that the spacing between the neighboring atomic layers of the optimized surface structure changes by less than 5% from the original spacing. The kinetic energy cutoff was set to 400 eV for all subsequent VASP calculations. The 2 × 4 × 3 and 3 × 4 × 1 Γ-point centered Monkhorst–Pack meshes generated using the VASPKIT 1.3.5 software package [31] were used for bulk and (100) surface of BCC AlCrTiNiV, respectively. The maximum forces and energy changes on each atom were converged to less than 0.02 eV/Å, and 1 × 10^−6^ eV/atom, respectively. The lattice constant of BCC AlCrTiNiV was determined to be 3.0 Å.

H_2_ dissociation, diffusion, and penetration processes were studied using TST. The Arrhenius expression for a chemical rate under the harmonic transition state theory was employed to determine the relationship between the chemical reaction rate constant *ν* [18] and the activation energies and prefactors of hydrogen atom diffusion or H_2_ molecular dissociation processes:(1)$$\nu =\nu_{o}\exp(-\Delta E/k_{B}T)$$

Δ*E* is the energy difference between the transition state (TS) and the initial state (IS) and is also called the activation energy, while *T* is temperature and *k_B_* is the Boltzmann constant. Under classical limitations, *ν_o_* is the temperature-independent prefactor [19], which can be expressed as follows:(2)$$\nu_{0}^{cl}=\frac{\prod_{i=1}^{3N}\nu_{i}}{\prod_{j=1}^{3N-1}\nu_{j}^{*}}$$

Here, *N* is the number of relaxed vibrating atoms, and *ν_i_* and *ν_j_^*^* are the real vibrational frequencies associated with the IS and TS, respectively. The classical expression of the chemical reaction rate is as follows:(3)$$\nu_{cl}=\frac{\prod_{i=1}^{3N}\nu_{i}}{\prod_{j=1}^{3N-1}\nu_{j}^{*}}\exp(-\Delta E/k_{B}T)$$

If we want to account for quantum mechanical effect, a zero-point energy (*ZPE*) correction must be added to the classical energy. This results in the quantum mechanical expression for diffusion rate under the harmonic transition state theory being
(4)$$\nu_{qm}=\frac{\prod_{i}^{3N}\nu_{i}f\left(h\nu_{i}/2k_{B}T\right)}{\prod_{j}^{3N-1}\nu_{j}^{*}f\left(h\nu_{j}^{*}/2k_{B}T\right)}\exp[(-\Delta E-\Delta E_{ZPE})\frac{1}{k_{B}T}]$$
where *f(x) = sinh(x)/x*. At high temperatures, *f(hν/2k_B_T) →* 1, leading to the quantum-corrected diffusion rate being simplified to classical expression we proposed above. At very low temperatures, the limiting expression is even simpler [32]:(5)$$\nu_{qm}=\frac{k_{B}T}{h}\exp(-\frac{\Delta E+\Delta E_{ZPE}}{k_{B}T})$$
(6)$$\Delta E_{ZPE}=\sum_{j}^{3N-1}\frac{h\nu_{j}^{*}}{2}-\sum_{i}^{3N}\frac{h\nu_{i}}{2}$$

Δ*E_ZPE_* is the difference between zero-point energies (*ZPE*) of the TS and IS. Additionally, we have to pay attention to a crossover temperature, *T_C_*, defined by
(7)Tc=hνIm*(ΔE+ΔEZPE)/kB2π(ΔE+ΔEZPE)−hνIm*ln2

Below this crossover temperature, quantum tunneling can make a significant contribution to the total diffusion rate, but otherwise it can be ignored [33]. By comparing the classical limit expression with quantum mechanical correction, we can conclude that the prefactor is temperature-independent in classical expression, depending only on frequency, but temperature-dependent in quantum expression. Additionally in the quantum expression, the zero-point energy correction can alter the activation energy.

The nudged elastic band (NEB) method is widely used for finding transition states in plane-wave DFT calculations. It was developed by Hannes Jόnsson and co-workers as a refinement of earlier “chain-of-states” methods that define the minimum energy path (MEP) between two local minima [34]. Later, an improved method called the climbing image-nudged elastic band (CINEB) method [35] was proposed, which is more accurate and stable than the NEB method. We needed to ensure that the initial state and final state are located at energy minima before performing the CINEB calculation. The initial estimated MEP should be as close as possible to the true path, as this is the most important factor influencing the convergence rate since CINEB is an iterative minimization method. One of the images was made to climb along the elastic band to rigorously converge on the highest saddle point. The calculation can converge when the forces acting on every atom are less than 0.02 eV/Å. The metal atoms were kept fixed during the frequency calculation. If there is only one imaginary frequency at the saddle point, we can confirm that the saddle point is the transition state. ZPE correction was obtained by summing the vibrational energies corresponding to the real vibrational frequencies of hydrogen atom.

Adsorption energy is defined as follows:(8)$$\Delta E_{ads}=E_{\left[slab+adsorbate\right]}-(E_{\left[slab\right]}+E_{\left[adsorbate\right]})$$

*E*_[*slab+adsorbate*]_, *E*_[*slab*]_, and *E*_[*adsorbate*]_ refer to the calculated total energy of the adsorbate on the AlCrTiNiV (100) surface, a clean AlCrTiNiV (100) surface, and a gas-phase H_2_ molecule or H atom in free space, respectively. Negative value of Δ*E_ads_* indicates that adsorption is more stable than the sum of the corresponding clean surface and isolated H atoms or H_2_ molecules.

## 3. Results and Discussion

### 3.1. Dissociation of H_2_ on AlCrTiNiV (100) Surface

A (100) surface of (3 × 2) BCC high-entropy alloy AlCrTiNiV, composed of 10 atomic layers and a total of 60 atoms in a bulk model, was constructed for surface research. After standard relaxation calculations, the variation of the atomic coordinates was small enough to indicate that the model is stable. The percentage change in the lattice spacing was less than 1% between the fifth and the sixth layer, suggesting that the fixed-layers section can reflect the internal characteristics of the bulk. Therefore, a slab model of the (100) surface with 60 atoms is sufficiently large for further calculations.

For molecular H_2_ adsorption, both end-on (the H-H bond is perpendicular to the AlCrTiNiV (100) surface) and side-on (the H-H bond is parallel to the AlCrTiNiV (100) surface) configurations at the top site of the Ti atom were studied. The parallel configuration was confirmed to be structurally stable and the atomic hydrogen adsorption energy was −0.15 eV/atom in the configuration. On the contrary, the vertical configuration is considered dynamically unstable because no steady atomic adsorption state can be obtained after convergence near the initial structure. Hence, the former was used as the initial state for the H_2_ dissociation study. For atomic H adsorption, different adsorption sites, including the top site, bridge site, and hollow site were all calculated, compared, and analyzed. The adsorption sites, adsorption energies, and vertical surface distances are listed in Table 1. We can conclude that the hollow site is the most stable site because the hydrogen atomic adsorption energy at the hollow site is the lowest compared to both the top site and the bridge site. Hence, two hydrogen atoms are adsorbed at the two hollow sites, which represent the most stable atomic adsorption state and this configuration was considered the final state for the H_2_ dissociation study. The dissociation energy barrier and dissociation pathway are plotted in Figure 2. The image shows that, in the transition state, the two hydrogen atoms are located at the bridge site and the activation energy of 0.11 eV is low enough for hydrogen atoms to be easily acquired. In other words, it is easy for the H_2_ molecule to dissociate and adsorb on the surface. Because of the reversibility of thermal motion at the microscopic level, the process of hydrogen recombination is the reverse of hydrogen dissociation. According to Figure 2g, the activation energy of the reverse process is 1.31 eV, which is large enough to prevent hydrogen atoms from recombining into hydrogen molecules. Overall, it is an exothermic process, which is different from the reverse endothermic process with a higher energy barrier, indicating that H_2_ dissociation on the surface is significantly easier than hydrogen atom recombination. 

The H-H bond length in the initial state was calculated to be 0.80 Å, which is slightly larger than 0.75 Å in the gas phase, using the same calculation parameters as in this study. The difference is caused by the forces exerted by the metal atoms on the surface and the closer the hydrogen atoms are to the surface, the greater the distance between the two hydrogen atoms. The distance between the H_2_ molecule and the surface (Ti atom) is 1.99 Å, as is shown in Table 1. In the transition state, the H-H bond length is 0.92 Å, and the distance between the midpoint of the two hydrogen atoms located in the bridge site of the two Ti atoms and the midpoint of the two Ti atoms is 1.42 Å, which approximately represents the distance of the two hydrogen atoms to the surface. In the final state, the two hydrogen atoms are located in the two adjacent hollow sites and the distance between the two is 2.30 Å greater than the distances of the first two states. The height differences between the two hydrogen atoms and the central points of the four surface atoms nearest to each hydrogen atom are 0.42 Å and 0.59 Å, respectively, which is the nearest of all the three states.

In reality, the concentration of the dissociated hydrogen atoms on the surface and the ambient temperature can influence the atomic adsorption energy and diffusion rate of hydrogen. The H_2_ dissociation process and its characteristics are described as follows.

The H_2_ molecule can be viewed as a rigid body, which is different from two hydrogen atoms, hence, the initial state of the H_2_ dissociation process needs to be analyzed separately. The linear H_2_ molecule has one vibrational, three translational, and two rotational degrees of freedom. Among them, the ZPE comes from the contribution of the vibration energy, as well as the rotational and translational parts, which depend on the pressure and temperature [36]. The H_2_ molecule has a vibrational frequency of ν_H2_ = 3501 cm^−1^, corresponding to a ZPE correction of 0.217 eV for the parallel site. It is noteworthy that there is an interaction force between the H_2_ molecule and the atoms on the surface, in contrast to the free H_2_ in the gas phase, as confirmed by our earlier analysis, resulting in a lower ZPE. The dissociation rate of the H_2_ molecule in the quantum mechanical expression can then be written as follows:$$r_{qm}=\frac{2h^{2}\varepsilon_{rot}P}{kT{\left(2\pi mkT\right)}^{3/2}}\frac{\exp[-(\Delta E+\Delta E_{ZPE})/kT]}{\prod_{j}^{3N-1}\left[1-\exp(\frac{-h\nu_{j}^{*}}{kT})\right]}$$

Here, *ε_ro_* = 7.55 meV is the rotational constant for H_2_, *P* is the pressure of the gaseous H_2_, *T* is the temperature and m is the mass of a H_2_ molecule [37]. At the saddle point, five real frequencies and one imaginary frequency indicate that the saddle point is the real transition state. In addition, the ZPE correction increases the classical dissociation energy barrier by about 0.1 eV. As is shown in Figure 3, the dissociation rates at different temperatures and H_2_ pressures are described by tracing points. The H_2_ dissociation rate on the AlCrTiNiV (100) surface increases rapidly as the hydrogen pressure and temperature increase. Moreover, Δ*E* values of 0.5 eV and 1.0 eV were input into the formula with 300 K, 100 kPa, and other quantities held constant, yielding the values of 9.43 × 10^−2^ s^−1^ and 4.59 × 10^−10^ s^−1^, which are far less than the actual dissociation rate of 3.60 × 10^5^ s^−1^, suggesting that the activation energy is the main factor influencing the dissociation rate. Hence, we can conclude that the temperature and energy barrier are the primary factors affecting the dissociation rate of H_2_ on the AlCrTiNiV (100) surface.

### 3.2. Adsorption of Atomic H on AlCrTiNiV (100) Surface

Hydrogen atoms that dissociate from H_2_ molecules can be adsorbed on the surface. Five different adsorption sites, including top sites, bridge sites, and hollow sites, were identified as potential sites after optimization. Adsorption energies, optimized adsorption sites, and the approximate vertical distances between the surface, for a single H atom, are shown in Table 2. As can be seen from the table, the most stable sites of the three types are the hollow sites, while the top site is the least likely to be adsorbed based on a comparison of the adsorption energies. Frequency calculations were also performed to confirm that all of the five sites are the real energy minimum points without any imaginary frequencies. Among the three hollow sites, the one with the lowest adsorption energy and the shortest distance of 0.5 Å to the surface is the most stable adsorption site. Moreover, the adsorption energy at all six of the top sites was calculated and compared with the fixed x and y coordinates as is shown in Table 3. Only the Al-top site, with three real frequencies, is dynamically stable, whereas the other sites are unstable with at least one imaginary frequency.

Next, we also studied the influence of coverage on the surface adsorption ability. The coverage of the hydrogen atoms was divided into six categories including the 1/6 monolayer (ML), 1/3 ML, 1/2 ML, 2/3 ML, and 1ML in this study. The coverage of the 1ML is defined as one H atom per top site on the surface. As can be seen in Table 3, different coverages correspond to the adsorption energies of −0.818 eV/atom, −0.723 eV/atom, −0.735 eV/atom, −0.655 eV/atom, and −0.534 eV/atom at 1/6 ML, 1/3 ML, 1/2 ML, 2/3ML, and 1ML, respectively. The adsorption energy tends to increase with increasing H coverage, indicating that the adsorption capacity is decreasing. This is caused by the lateral repulsions between the hydrogen atoms, which causes the hydrogen atoms to deviate from their original sites. The lattice structures are confirmed to still be dynamically stable, without imaginary frequencies. Moreover, ZPE corrections can make surface adsorption more difficult. The higher the coverage, the shorter the distance between the hydrogen atoms, contributing to the hydrogen atoms attracting each other, forming H_2_ molecules, and keeping the H_2_ molecules far away from the surface, which further prevents the surface from adsorbing new H atoms.

### 3.3. Diffusion of Atomic H on the Surface and Penetration into the Bulk

The diffusion process of the atomic H on the surface was investigated using the diffusion path from the 4′ site as the initial state and from the 3 site as the final state. The energy barrier is shown in Figure 4, with an activity energy of 0.41 eV. The hydrogen atoms at the saddle point are located at the bridge site. By performing a frequency calculation, three real frequencies of H at the ground state, and two real frequencies along with one imaginary frequency at saddle point, were obtained, indicating that the bridge site is a real transition state. To compare this result with those in other reports, another form of the diffusion rate in the classical limitation [38] was also applied, which is as follows:$$D_{0}^{cl}=\frac{1}{6}\lambda^{2}\frac{\prod_{i=1}^{3N}\nu_{i}}{\prod_{j=1}^{3N-1}\nu_{j}^{*}}$$

Here, *λ* is the hopping distance of the hydrogen atom between the initial state and the final state. We then determined that the pre-exponential factor of the diffusion rate is *D_*0*_^cl^* = 6.13 ps^−1^ or *D_*0*_^cl^* = 2.25 × 10^−7^ m^2^/s in the classical limit. The ZPE corrections applied to the activation energy and the quantum-mechanical pre-exponential factors *D_*0*_^qm^* at different temperatures for various diffusion paths are shown in Table 4.

We further analyzed the diffusion of H atoms through the subsurface and their subsequent penetration into the bulk. Figure 5 shows the barriers for H penetration from the surface to the bulk. The entire permeation process can be divided into three smaller processes. H diffusion from the B site to the C site involves a significantly endothermic energy barrier of 0.856 eV, compared with the barrier of 0.148 eV for H diffusion from the A site to the B site, indicating that it is more difficult for H to diffuse below the surface atomic layer than above it. The activity energy between the C and G sites is smaller than the barrier energy from the B site to the C site, which could be caused by the severe lattice distortion of the top five atomic layers that are relaxed. Lattice distortion plays a significant role in influencing hydrogen diffusion in HEAs by altering diffusion paths, creating trapping sites, and inducing anisotropic diffusion behavior. As shown in this alloy, this hydrogen atom diffusion path is not as regular as the one in the VC without the lattice distortion effect [18]. The activity energy between the G and K sites in the bottom five atomic layers, where the atoms are arranged in reverse order, becomes greater, which verifies the point mentioned above. Overall, at least 2.057 eV of energy is required for a hydrogen atom to penetrate into the bulk. A frequency calculation was then performed to determine the pre-exponential factor. Three real frequencies at the ground state, and two real frequencies and one imaginary frequency at the saddle point, were found for all the H diffusion steps from the A site to the K site. Nearly all of the ZPE corrections to the penetration energy barriers were negative, making penetration easier. The corresponding pre-exponential factor in the classical limitation, D_0_^cl^, and the pre-exponential factor, D_0_^qm^, with the quantum correction of H as a function of temperature, are shown in Table 4. The pre-exponential factor D_0_^qm^ tends to increase with rising temperature, and slowly approach the pre-exponential factor in the classical limit, D_0_^cl^. This can be explained mathematically by the expression ν ≈ kT(1 − e^hν/kT^)/h as hν/kT approaches 0, indicating that as the temperature increases, the two values converge. This shows that the quantum mechanical effect and elevated temperatures facilitate hydrogen atom permeation.

In addition, H atoms are preferentially trapped in the tetrahedral interstitial sites and penetrate through the atomic layer between the two nearest tetrahedral interstitial sites. The height of the barriers depends on the number of H-M band making/breaking events. Overall, the highest energy barrier of 0.844 eV occurs during the diffusion step from the surface layer to the subsurface layer.

Compared with pure metals that have similar BCC structures, such as Fe with a barrier of 0.088 eV and a pre-factor of 4.212 × 10^−8^ m^2^/s [39], H atoms need to overcome higher dissociation, diffusion, and penetration energy barriers with pre-factors of the same order of magnitude for AlCrTiNiV, even without considering the ZPE corrections. This indicates that the AlCrTiNiV (100) surface can prevent H from permeating and passing through the material because of this high-activation energy barrier.

On the basis of the theoretical results of this paper, for the next step, we will investigate its hydrogen permeability through experiments. Specifically, the high-entropy alloy film will be designed and fabricated through atomic layer deposition, and then the hydrogen permeability of the film will be tested through experiments. Furthermore, by substitution of the elements in the high-entropy alloy film, the hydrogen permeability will be optimized, finally finding the alloy components with better hydrogen resistance performance than those in the current market, such as Al_2_O_3_, VC, etc.

## 4. Conclusions

The DFT was used to study the diffusion behavior of hydrogen in a new type of alloy material called high-entropy alloy (HEA). The different atomic sizes cause [18] the structure to lose its perfect symmetry and can even produce severe lattice distortion in some areas, varying hydrogen’s preferred diffusion sites and paths. The pre-factor was determined by performing vibrational frequency calculations according to TST, with quantum corrections under different temperature conditions taken into account. Detailed theoretical calculations and analysis of the dissociation, diffusion, and penetration of H_2_/H show that the main resistance to H permeation occurs at the surface layer. In the bulk, H at locations with less lattice distortion tends to encounter higher energy barriers. Areas where more H-M bonds are breaking and forming correspond to higher diffusion barriers. The fundamental physics mechanism of AlCrTiNiV as a hydrogen-permeation barrier (HPB) is the strong electrostatic interactions between hydrogen and metal atoms, forming covalent bonds during diffusion.

## Figures and Tables

**Figure 1 nanomaterials-14-01391-f001:**
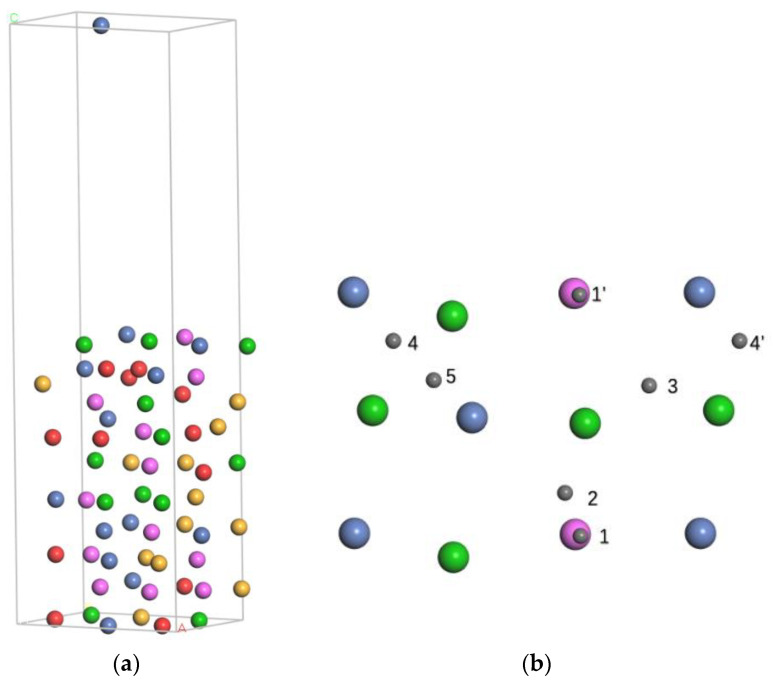
(**a**) AlCrTiNiV (100) surface after relaxion with 10 atomic layers and vacuum space of 15 Å. The red ball is V atom, green is Ti atom, pink is Al atom, yellow is Cr atom and blue is Ni atom. (**b**) Single hydrogen atom adsorption sites on the AlCrTiNiV (100) surface. Sites 1 to 5 are the top site of Al atom, Al-Ti bridge site, three different fourfold hollow sites, respectively. The 1’ site and 4’ site are the same sites as the 1 site and 4 site, respectively. The grey ball is H atom.

**Figure 2 nanomaterials-14-01391-f002:**
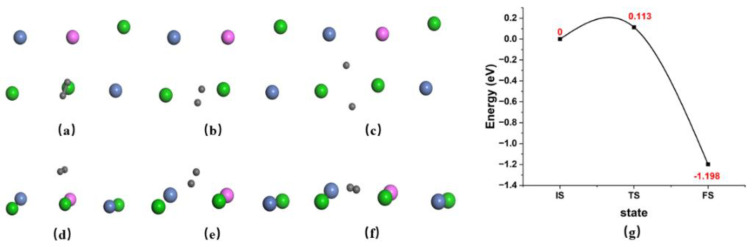
(**a**–**c**) The top view of the initial state, transition state, and final state. (**d**–**f**) The front view of the initial state, transition state, and final state. The green ball is Ti atom, pink is Al atom, and blue is Ni atom. (**g**) The dissociation pathway of $H_{2}$ molecule from top site of Ti atom to the hollow sites via the bridge site with the energy barrier of 0.113 eV.

**Figure 3 nanomaterials-14-01391-f003:**
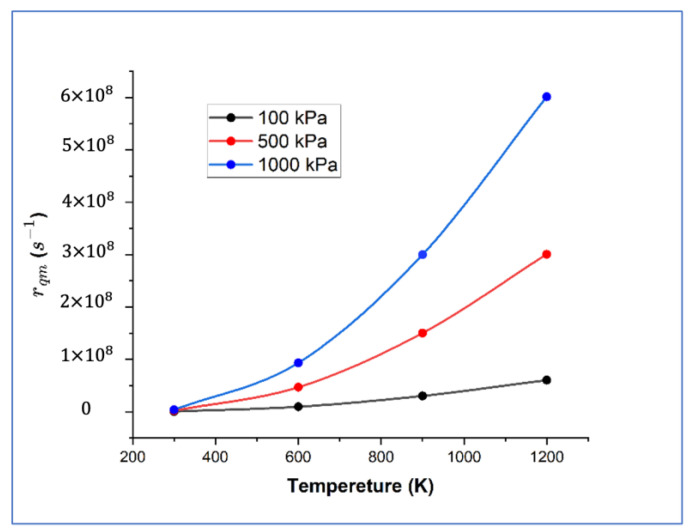
Dissociation rates at different temperatures and H_2_ pressure conditions.

**Figure 4 nanomaterials-14-01391-f004:**
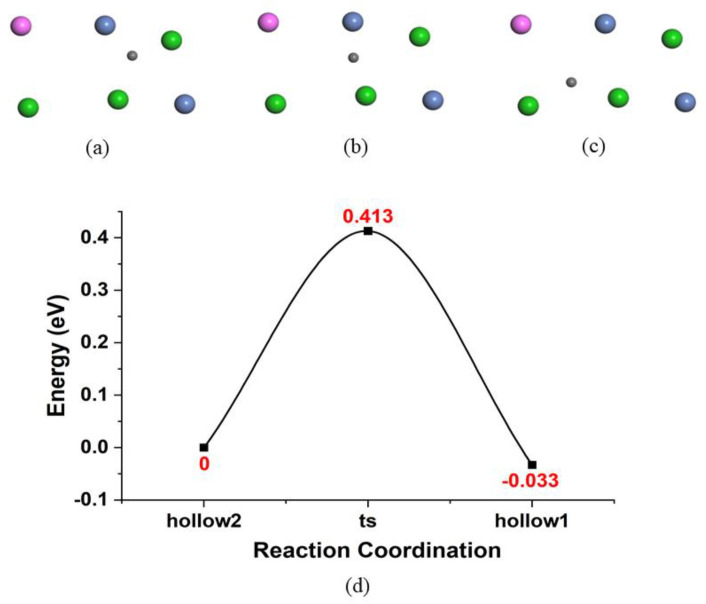
(**a**–**c**) The top view of the initial state, transition state, and final state. The green ball is Ti atom, pink is Al atom, and blue is Ni atom. (**d**) The diffusion pathway of a hydrogen atom from the 4′ site to the 3 site, via the bridge site, with an energy barrier of 0.413 eV.

**Figure 5 nanomaterials-14-01391-f005:**
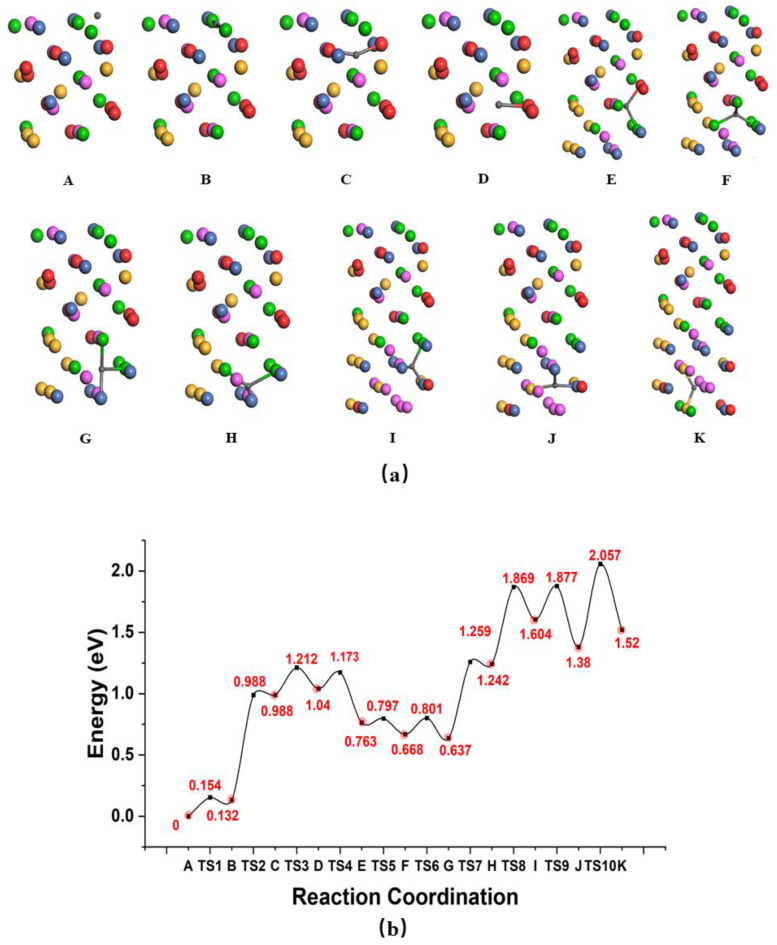
(**a**) A to K correspond to continuous diffusion structures. The red ball is V atom, green is Ti atom, pink is Al atom, yellow is Cr atom and blue is Ni atom. (**b**) The barriers of H penetration from surface to bulk AlCrTiNiV. H diffusion from B site to C site, with the largest of all the diffusion barriers, and from G site to K site, with relatively large barriers, shows that the main resistance effect of H penetration may take place at the surface layer and bulk area.

**Table 1 nanomaterials-14-01391-t001:** The adsorption energy and distance to surface at different adsorption sites on surface.

Adsorption Site		$\Delta E_{ads}\;(eV/atom)$	Distance to Surface (Å)
H_2_ molecule	Ti-Top	−0.152	1.994
H atom	1 (Top)	−0.123	1.590
	2 (Bridge)	−0.463	1.109
	3 (Hollow)	−0.863	0.597
	4 (Hollow)	−0.830	1.035
	5 (Hollow)	−0.803	1.040
2 H atoms	Hollow	−0.723	0.418, 0.587

**Table 2 nanomaterials-14-01391-t002:** Calculated Energy, frequency, and band-length of H-M at different top sites by fixing the x and y coordinates.

Adsorption Site		Top-Al1	Top-Ti1	Top-Ti2	Top-Ti3	Top-Ni1	Top-Ni2
Energy (eV)		−421.72	−421.30	−421.25	−421.33	−421.81	−421.54
Frequency (THz)	1	51.36	46.10	44.31	43.86	52.53	52.37
	2	11.74	6.40	1.12i	8.09i	3.89i	2.22i
	3	5.52	3.37i	10.46i	11.49i	13.60i	8.46i
Band-length of H-M (Å)		1.59	1.77	1.78	1.77	1.53	1.5

**Table 3 nanomaterials-14-01391-t003:** Calculated energy, adsorption energy, and adsorption energy corrected by zero-point energy of different coverages.

Coverage (ML)	Energy (eV)	E_ad_ (eV/atom)	E_ad_^ZPE^ (eV/atom)
1/6	−422.46	−0.86	−0.82
1/3	−426.49	−0.75	−0.72
1/2	−430.67	−0.77	−0.74
2/3	−434.48	−0.68	−0.65
1	−441.87	−0.56	−0.53

**Table 4 nanomaterials-14-01391-t004:** Hydrogen atom diffusion on the AlCrTiNiV (100) surface, and penetration through the surface and bulk: Energy barriers, ZPE corrections, corresponding calculated pre-factors D_0_^qm^ at different temperatures, the classical pre-factors D_0_^cl^ in different expressions, and crossover temperature T_C_.

Path		ΔE (eV)	ΔZPE (eV)	D_0_^qm^ (ps^−1^)			D_0_^cl^ (ps^−1^)	D_0_^cl^ (1 × 10^−8^ m^2^s^−1^)	T_C_ (K)
				T = 300 K	T = 600 K	T = 900 K			
Diffusion on surface		0.431	−0.034	6.128	10.512	12.996	20.16	22.487	116
Penetration from surface	A→B	0.148	−0.027	6	9.83	11.85	17.07	3.449	70
	B→C	0.856	−0.011	6.11	10.45	12.75	17	8.169	61
	C→D	0.224	−0.03	5.91	9.16	10.7	15.17	18.024	151
	D→E	0.132	−0.003	5.542	7.721	8.45	9.363	4.233	100
	E→F	0.033	−0.043	6.123	10.654	13.419	16.89	3.139	126
	F→G	0.132	−0.018	6.157	10.688	13.148	18.343	4.235	139
	G→H	0.621	0.016	6.07	10.04	11.81	12.721	4.853	159
	H→I	0.625	−0.001	6.118	10.424	12.627	15.729	3.632	251
	I→J	0.271	−0.03	6.223	11.515	14.981	24.466	7.267	236
	J→K	0.6	0.056	6.027	9.354	10.239	7.856	4.433	231

## Data Availability

Data are contained within the article.

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
