# Peer review of "Mechanism for Adsorption, Dissociation, and Diffusion of Hydrogen in High-Entropy Alloy AlCrTiNiV: First-Principles Calculation"

_nanomaterials, 2024, doi:10.3390/nano14171391_

Round 1

Reviewer 1 Report

Comments and Suggestions for Authors

This manuscript entitled " Mechanism for adsorption, dissociation and diffusion of hydrogen in hy-drogen permeation barrier of high-entropy alloy AlCrTiNiV: The first principle theory calculation” written by Weilong Zheng and co is well presented and described, However, before its publication, I would like to suggest some minor revision that should be addressed.

1.      Can you elaborate on the choice of the high-entropy alloy (HEA) AlCrTiNiV for this study? What specific properties of this alloy make it suitable for investigating hydrogen permeation barriers?

2.      The study uses density functional theory (DFT) and transition state theory (TST) to model hydrogen behavior. How do these theoretical approaches complement each other in understanding hydrogen adsorption, dissociation, and diffusion in HEAs?

3.      The article mentions the potential impact of lattice distortion on the hydrogen permeation rate. Can you provide more details on how lattice distortion influences hydrogen diffusion and the methods used to quantify this effect?

4.      In the results section, you discuss the adsorption energy of hydrogen on different atomic sites. How do the adsorption energies at these sites compare, and what implications do these differences have for the design of effective hydrogen permeation barriers?

5.       The paper concludes that high-entropy alloys have the potential as hydrogen-resistant coating materials. Based on your findings, what are the next steps for experimental validation, and how might this research be applied in practical settings?

Reviewer 2 Report

Comments and Suggestions for Authors

The manuscript describes a computational study of hydrogen adsorption and diffusion within an AlCrTiNiV high-entropy alloy. High-entropy alloys are currently the focus of substantial research effort, in view of their unique structural, mechanical and electronic properties. Because of their thermodynamic stability to hydrogen permeation, they can provide suitable support materials for hydrogen storage. The goal of this paper is to analyze how a representative model high-entropy alloy surface adsorbs and favors the dissociation of hydrogen molecules, and how the structure of the alloy influences the diffusion of the resulting H atoms in the alloy bulk. The authors use static density-functional theory (DFT) calculations on a periodic sample of AlCrTiNiV, along with transition state theory and the climbing image nudged elastic band method, to study the H2  adsorption and dissociation and to model the migration of the H atoms between specific sites in the crystal structure of the alloy. The results indicate that H2 adsorption occurs preferentially with H-H bonds parallel to the alloy's surface and that this is an exothermic phenomenon. The diffusion of H atoms is influenced by the specific geometry of the sites visited by the H atoms along their diffusion path. Locations with reduced structural distortion are shown to exhibit higher diffusion barriers, in consequence to the larger number of metal-H bonds that need to be broken to allow the H atom to migrate. Quantum effects (zero point energy correction) are also shown to promote H migration.

This study addresses a very topical subject of both fundamental and applied technical interest. The goals of the work and the research design are clear. The methods used are appropriate and sound. The results are, in general, convincing. However, I think there are various issues that should be addressed to make the study more convincing and useful for further work on the subject.

1) The theoretical approach described in section 2 for modeling H2 dissociation and H diffusion relies on the harmonic approximation. This seems to be a very drastic approximation in this case, considering that the H-H bond is being broken, and new metal-H bonds are formed and broken during the diffusion of H in the bulk alloy. How do the authors justify the use of the harmonic approximation? Can the results be strengthen using non-harmonic corrections and/or ab initio molecular dynamics simulations (which explicitly account for anharmonicity)?

2) On page 5, the authors show that the H2 molecule tends to adsorb on the alloy surface with its H-H bond parallel to the surface, rather than head-on. What causes this preferential arrangement? Considering the finding described on page 6, concerning the lengthening of the H-H bond length compared to the free molecule, this is likely to be caused by electron transfer either from metal atoms to the anti-bonding sigma* orbital of H2 or from the bonding sigma orbital of H2 to empty bands of the alloy. Can this be verified, e.g. by computing Mulliken (or other kinds of) partial atomic charges? This is an important point, because electron transfer from/to H2 is the driving force of the molecular dissociation upon adsorption.

3) The authors carried out all calculations using PBE. Although, in general, this functional can provide a satisfactory description of structural and vibrational properties of extended systems, it may fail to give a quantitative account of the energies of the molecular orbitals of H2 relative to the extended bands of the alloy, largely because of errors caused by self-interaction. It may be worth verifying that the results concerning the H2  adsorption and the initial stages of the dissociation (including the H-H bond lengthening) are similarly described by more advanced exchange-correlation functional approximations for DFT, e.g. global or range-separated hybrids. 

4) As mention in section 2, the DFT calculations were spin-polarized. What was the total spin moment, and how did it evolve during the H2 dissociation?  As the H-H bond breaks during the dissociation, a triplet state should form. However, the electron on each of the incipient H atoms can couple with unpaired electrons on the Ni atoms. Was this phenomenon observed in the simulations, or did the spin change drastically upon dissociation?

5) Figure 2(g) shows that the dissociation of H2 on the alloy surface is exothermic. This implies that, thermodynamically, an H2 molecule will always remain dissociated into atoms when in contact with, or diffusing within, the alloy. Through what mechanism could then H2 be released after diffusing?

6) The quality of the presentation should be improved substantially. There are several text imprecisions and errors, which often render the description unclear and confusing. The links to the references in the main text are also poorly formatted. For instance, on line 30, "1-4" appears as "1234". The title of the paper, also, should be improved. I recommend changing it to "Mechanism for adsorption, dissociation and diffusion of hydrogen in the high-entropy alloy AlCrTiNiV: First-principles calculations".

Comments on the Quality of English Language

As mentioned at point 6 in the Comments and Suggestions for Authors, the text requires substantial improvements. There are several typos, errors, imprecisions and unclear sentences that require attention. 

Round 2

Reviewer 2 Report

Comments and Suggestions for Authors

The authors have addressed satisfactorily the points raised in my previous report and they have improved the overall quality of the presentation. I believe the manuscript is now acceptable for publication after minor text adjustments and spell checking. 

Comments on the Quality of English Language

In the revised manuscript, the authors have addressed the majority of the text/language issues of the original submission. The paper is now readable and clear. Minor text changes and spell checking may still be required before publication.